# Autosomal Recessive Limb-Girdle Muscular Dystrophy-3: A Case Report of a Patient with Autism Spectrum Disorder

**DOI:** 10.3390/genes14081587

**Published:** 2023-08-05

**Authors:** Sivan Lewis, Amy Woroch, Mary Kate Hatch, Reymundo Lozano

**Affiliations:** 1Department of Genetics and Genomic Sciences, Icahn School of Medicine at Mount Sinai, New York, NY 10029, USAreymundo.lozano@mssm.edu (R.L.); 2Department of Pediatrics, Icahn School of Medicine at Mount Sinai, New York, NY 10029, USA

**Keywords:** autism spectrum disorder, developmental delay, limb-girdle muscular dystrophy

## Abstract

Limb-girdle muscular dystrophies are a group of genetic disorders classically manifesting with progressive proximal muscle weakness. Affected individuals present with atrophy and weakness of the muscles of the shoulders and hips, and in some cases, intellectual disability or developmental delay has also been reported. Limb-girdle muscular dystrophy-3 is a recessive disorder caused by biallelic variants in the *SGCA* gene. Similarly, symptoms include proximal muscle weakness, elevated CPK, calf muscle pseudohypertrophy, and mobility issues. Cardiac symptoms and respiratory insufficiency are also common symptoms. This case report details a 3-year-old male with muscular weakness, elevated CK, and a neurodevelopmental disorder in whom a homozygous missense variant in c.229C>T (p.Arg77Cys) associated with limb-girdle muscular dystrophy-3 was found. This report shows the association between *SGCA* c.229C>T and neurodevelopmental disorders as observed in other muscular dystrophies.

## 1. Introduction

Limb-girdle muscular dystrophies (“LGMD”) encompass a diverse group of progressive disorders primarily affecting proximal muscles, with considerable variation in age of onset, progression, and severity. Due to genetic and phenotypic heterogeneity, the nomenclature of these conditions evolved over time, without clear consensus. The ENMC International Workshop in 2017 refined the classification to include 29 genetic subtypes, considering the pattern of inheritance and proximal and distal-proximal muscle involvement, and incorporating criteria such as muscle fiber degeneration, elevated CK levels, and MRI findings indicative of degenerative changes in individuals capable of independent walking [1,2]. The proposed naming formula was ‘LGMD, inheritance (R or D), order of discovery (number), affected protein.’ Under this formula, LGMD2D was renamed LGMDR3, α-sarcoglycan-related.

Autosomal recessive limb-girdle muscular dystrophy-3 (“LGMDR3”) is a recessive disorder caused by biallelic pathogenic variants in the *SGCA* gene which is the α subunit of the sarcoglycan adhalin. Sarcoglycans are glycoproteins that interact with the dystrophin-glycoprotein complex (DGC) which maintains the integrity of cell membranes as myocytes relax and contract. Mutations in the *SGCA* gene disrupt this process and manifest with a range of neuromuscular phenotypes [3].

LGMD is generally rare. There are few reports on its prevalence due to heterogeneity and evolving nomenclature, with an estimated prevalence from 1/14,500 to 1/123,000 [4]. Among all LGMD patients, sarcoglycanopathies account for 10–25% of cases in most countries, while LGMDR3 is the most common of the sarcoglycanopathies. Additionally, consanguineous families have a higher frequency of LGMD [1]. In a large cohort of 208 Danish patients with LGMD phenotype, *SGCA* variants were found in 12 patients (5.7%) [5].

The symptoms of sarcoglycanopathies include elevated CPK, calf muscle pseudohypertrophy, positive Gowers’ sign, toe walking, and mobility issues, with varying degrees of cardiac and respiratory involvement. LGMDR3 has the most variable phenotype of the sarcoglycanopathies, with severe and mild presentations reported [1]. Approximately 10% of individuals with LGMDR3 will experience cardiac symptoms, primarily cardiomyopathy, and up to 27% of patients can suffer respiratory insufficiency and require ventilatory support [3]. The onset of symptoms occurs at the mean age of 10 years old.

The 2014 guidelines by the American Academy of Neurology for diagnosing and treating limb-girdle muscular dystrophies [6] outline a diagnostic approach that primarily prioritizes targeted genetic testing, drawn from unique features, patterns of weakness, and inheritance. If no diagnosis results from this testing, a muscle biopsy is proposed, and if still undiagnosed, next-generation sequencing (NGS) is offered as a final recourse. Notably, Yis et al. suggested that in the case of sarcoglycanopathies, any identified sarcoglycan defects on biopsy should be genetically confirmed [7].

As genetic testing has become more accessible and less costly, the utilization of NGS in LGMD diagnosis has expanded. Compared to traditional methods such as Sanger sequencing, biochemical, and histopathological approaches that had a diagnostic yield of 30–40% [8], the yield of exome sequencing for LGMD has shown considerable improvement, reaching the 40–45% range [9,10,11]. Custom target capture-based NGS panels have found a 33% overall diagnostic rate and detected pathogenic or likely pathogenic variants in 40% of patients with sarcoglycanopathies [12].

A recent study conducted by Winckler et al., which employed NGS panels as the first diagnostic approach for muscular dystrophies and myopathies, reported a promising 52.9% overall diagnostic rate and a 48% rate specifically for LGMD [13].

Presently, the diagnostic pathway may evolve to showcase NGS as the primary diagnostic option in certain cases, with a muscle biopsy suggested if NGS is inconclusive [14]. If a clear pathogenic mutation is identified via NGS, no further diagnostic work is required.

Importantly, NGS not only provides a promising leap in diagnostic yield but broadens our understanding of the wider implications of LGMD. This deeper comprehension is especially valuable when delving into under-investigated areas of association, such as the potential link between LGMD and neurodevelopmental disorders.

While neurodevelopmental disorders are associated with other neuromuscular disorders such as Duchenne or Becker muscular dystrophy [15], they have not been clearly associated with limb-girdle muscular dystrophies. As many congenital muscular dystrophies are allelic with subtypes of LGMDs, we have performed a review of the literature for patients with non-congenital LGMD phenotypes and neurodevelopmental features (Table 1). Bögershausen et al. reported two families with *TRAPPC11* variants associated with myopathy, moderate ID, infantile hyperkinetic movements, and ataxia [16]. In a cohort of eight patients with *GMPPB*-related limb-girdle muscular dystrophy, speech delay, autism, and learning disability were described in three patients [17].

In addition, Haberlova et al. reported on two sisters with *POMT1*-associated muscular dystrophy who were described with global developmental delays before the onset of limb-girdle muscle dystrophy symptoms in their early 30s [18].

This case report details the presentation of a patient with LGMDR3 and autism spectrum disorder.

## 2. Case Presentation

A 3-year-old male with speech delay and suspected autism spectrum disorder presented with a 1-month history of weakness and elevated CPK (26,123 U/L). The patient was an only child, born via vaginal delivery at 41 weeks gestation to a 34-year-old mother who had an uneventful pregnancy and adequate prenatal care. Prenatal genetic evaluations included maternal cystic fibrosis, spinal muscular atrophy, and fragile X testing which were normal. Family history was negative for relatives with developmental delay, muscle weakness, or cardiomyopathy. Both parents originated from the same village in India and reported no known consanguinity (Figure 1).

The patient had reached appropriate developmental milestones by 2 years of age, when he developed speech regression. Audiology and ophthalmology examinations were normal. Suspicion of autism was raised for hand-flapping, decreased eye contact, nervousness around other children, and difficulty with transitions. He had unstable and weak knees and toe walking occasionally.

He had markedly elevated CPK (27,209 U/L), LDH (1346 U/L) and elevated liver enzymes (AST 422 U/L, ALT 386 U/L). Kidney function was normal, and urinalysis was normal. A liver ultrasound and autoimmune hepatitis workup were negative. The neurologic evaluation was significant for a slightly wide base gait and knees intermittently unsteady, with normal tone. Reflexes and gross motor strength assessment were limited due to poor cooperation. An echocardiogram was within normal measurements.

He was on the 40th and 34th percentile for his weight and height, respectively. He was non-verbal, mostly not cooperative, but consolable with his parents. He had a posteriorly rotated left ear and prominent ears with no other dysmorphic features. Cardiovascular and respiratory examinations were unremarkable. Mild calf pseudohypertrophy was noted bilaterally. When rising from a sitting position on the floor, he demonstrated a unilateral positive Gowers’ sign. An accurate assessment of strength was difficult because of poor cooperation due to autistic behavior; he appeared weak in the lower limbs with a wobbly and unstable gait. Neuropsychiatric evaluation by a developmental pediatrician revealed persistent deficits in communication with a vocabulary consisting of very few words, unintelligible speech, and frequent echolalia. Social interaction was deemed poor. He had restricted, repetitive patterns of behaviors, and stereotypic hand movements. He met the criteria for autism spectrum disorder (ASD) (Table 2). While he had symptoms associated with attention deficit hyperactivity disorder (ADHD), the patient did not meet the criteria for ADHD diagnosis.

A Comprehensive Muscular Dystrophy Panel was ordered with reflex whole-exome sequencing. Given his new diagnosis of autism spectrum disorder, karyotype and a chromosomal microarray were added to the evaluation.

The Comprehensive Muscular Dystrophy Panel identified homozygous pathogenic missense variants in the *SGCA* gene, namely c.229C>T (p.Arg77Cys) (Figure 2). This was consistent with a diagnosis of LGMDR3.

The chromosomal analysis revealed a normal male 46, XY karyotype, and chromosomal microarray revealed two regions of homozygosity on chromosomes 1 and 17; the *SGCA* variants identified in our patient were found within the region of homozygosity on chromosome 17. A heterozygous pathogenic variant and a heterozygous variant of uncertain significance were found in the *STXBP2* gene (c.1247-1 G>C;p.? and c.1286 C>T; p.A429V, respectively). The variants were in trans. Biallelic pathogenic variants in *STXBP2* cause autosomal recessive familial hemophagocytic lymphohistiocytosis type 5, which typically manifests with prolonged and high fevers, cytopenias, hepatosplenomegaly, and liver dysfunction in early childhood [19]. None of these symptoms were observed in our patient. Additionally, a maternally inherited hemizygous missense variant of uncertain significance was detected in the *SSR4* gene (c.7G>T; p.A3S). Loss of function pathogenic variants in *SSR4* are associated with X-linked SSR4-related congenital disorders of glycosylation. All males reported with this disorder had global developmental delays, intellectual disability, microcephaly, and typical facial dysmorphism; seizures and gastrointestinal disorders were also common [20]. Our patient did not have the typical facial features or microcephaly seen in these patients. No additional variants associated with developmental delay or autism were identified.

## 3. Discussion

Limb-girdle muscle dystrophy is a genetically heterogeneous disorder with significant phenotypic and genetic variability [1]. LGMDR3 is associated with the *SGCA* gene which encodes the α subunit of adhalin, a sarcoglycan critical in muscle tissue architecture through its stabilization of the dystrophin–glycoprotein complex (DGC).

Typically, patients with sarcoglycanopathies have a childhood onset of progressive limb-girdle atrophy and loss of ambulation [21], with varying degrees of cardiac and respiratory involvement. Toe walking in early childhood is often present before muscle weakness is detected. The mean age of onset in LGMDR3 is 10 years old [3]. Nevertheless, milder forms have been described and adult-onset weakness is especially seen in this type of LGMD [22].

Our patient had an early presentation with Gowers’ sign evident as early as 3 years of age. His unilateral presentation is also inconsistent with the symmetric muscle weakness usually described in these patients, but probably related to an early diagnosis. Our patient has one of the most common missense mutations in the *SGCA* gene, namely c.229C>T (p.Arg77Cys), which has been associated with a very reduced protein expression, and consequently with a more severe prognosis and rapid progression [3].

It has been established through tissue-specific expression analyses that the mRNA of adhalin is selectively restricted to striated muscle, in contrast to dystrophin mRNA which exhibits a broader expression profile including the brain. This finding has led to the postulation that the absence of *SGCA* gene expression in the brain may explain the lack of cognitive findings in patients with LGMDR3 as opposed to patients with dystrophinopathies [23,24]. The conjunction of muscle weakness with autism spectrum disorder in our patient suggests a possible wider phenotype associated with variants in *SGCA* and a diagnosis of LGMDR3 including behavioral and cognitive findings as seen in Table 2.

Muscle biopsy was not performed after diagnosis in our patient, given that the variants found were known as pathogenetic and in accordance with the guidelines set by the American Academy of Neurology [6]. As further data emerge regarding the increased diagnostic yield of NGS with LMGD, guidelines are likely to be updated to increase the uptake of genetic testing as the initial diagnostic workup.

While not required for diagnosis, the expression of α-sarcoglycan in muscle biopsies from individuals with LGMDR3 shows a relationship with disease onset. Severe cases of the condition demonstrate α-sarcoglycan deficiency. In contrast, in later-onset and milder cases [22], the muscle biopsies may have normal α-sarcoglycan immunohistochemistry.

Although a reduction in α-sarcoglycan can be used as a biomarker for clinical severity, it may not correlate with intellectual developmental disorders. A larger study of expression and clinical phenotype will be necessary to make an accurate determination of phenotype and genotype associations.

Extensive workup including cytogenetics and whole-exome sequencing did not reveal other genetic etiology for our patient’s autism, and the variants of uncertain significance described were not consistent with his phenotype. The workup did not include ACMG-secondary findings in the context of whole-exome sequencing, as the family declined citing the already overwhelming experience of receiving the patient’s diagnoses.

To date, there are no approved therapies for LGMDR3. Given its monogenic nature, several research groups are investigating gene therapy, with promising preliminary results. Mendell et al. conducted a Phase I/II trial involving isolated limb infusion of the *SGCA* human transgene via AAVrh74, an Adeno-associated virus vector, with enhanced muscle force but unaltered walk time [25]. Building on this method, Griffin et al. recently demonstrated the successful systemic delivery of the *SGCA* transgene in knockout mice [26], with improved locomotor activity and no detected toxicity. The potential of in vivo gene editing is also emerging as a means to prevent AAV-related toxicity. In an illustrative study by Escobar et al., the common *SGCA* c.157G>A mutation was successfully rectified via adenine base editing in two muscle stem cells [27], increasing α-sarcoglycan expression and muscle regeneration. A novel study in a murine model presents an alternative approach. This study demonstrated that small molecules belonging to the class of cystic fibrosis transmembrane conductance regulator (CFTR) correctors successfully rerouted α-SG to the sarcolemma, resulting in preserved muscle force; the authors propose that this method may prove simpler and more effective than complex gene or cell transfer [28].

The patient was referred to a research center for natural history research participation and potentially further clinical evaluation.

It must be acknowledged that we cannot fully exclude the possible contributions of other genetic factors to the observed neurodevelopmental pathology. Further research on larger cohorts with the c.229C>T variant focusing on development and behavioral aspects would be beneficial to better ascertain the significance of its correlation with these findings.

Such research could catalyze the provision of tailored counseling and developmental support services for affected patients.

## Figures and Tables

**Figure 1 genes-14-01587-f001:**
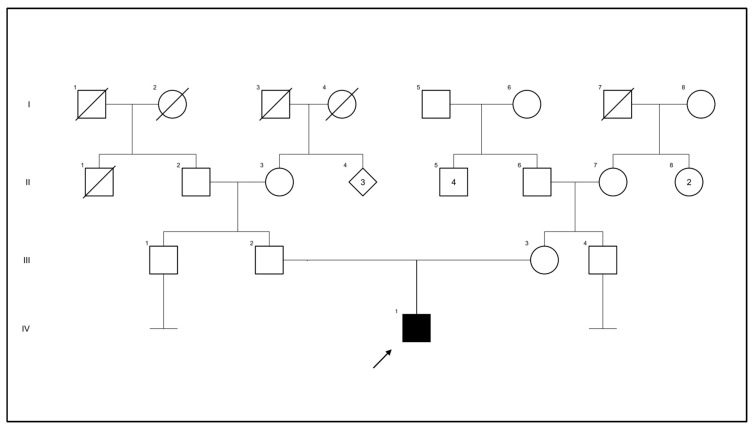
Pedigree of the family under study; proband indicated by the arrow. This pedigree diagram uses Roman numerals I–IV to represent generations, with I indicating the first generation and IV indicating the fourth. The numbers associated with individuals indicate their specific position within the respective generation.

**Figure 2 genes-14-01587-f002:**
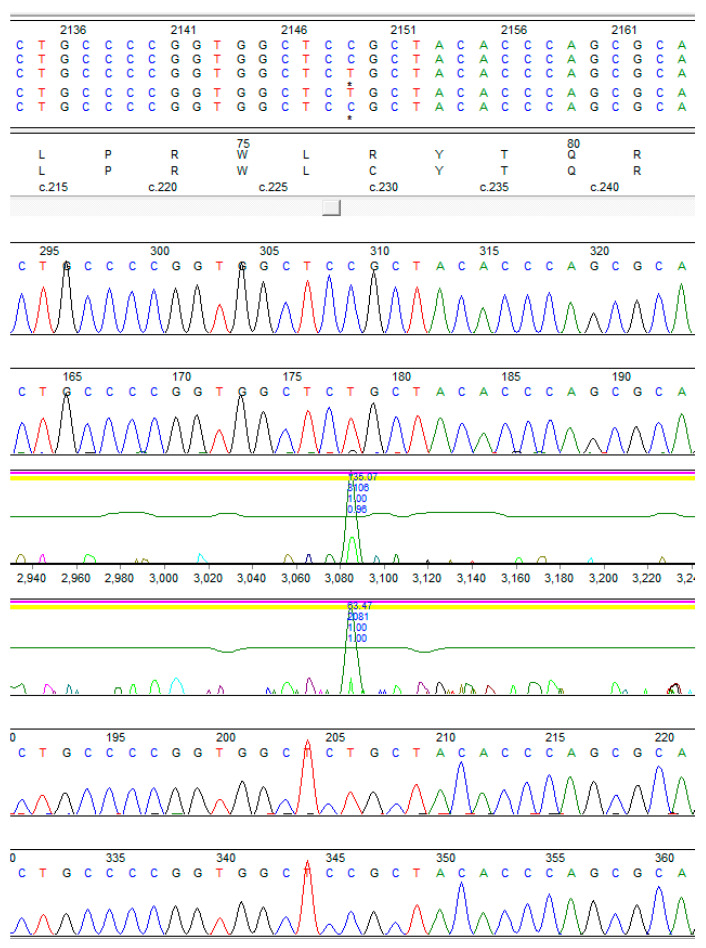
Sanger sequencing results: visual inspection of the sequencing chromatogram. The colors denote specific nucleotides-green for adenine (A), blue for cytosine (C), red for thymine (T), and black for guanine (G). A homozygous variant c.229 C>T, p.(R77C) was detected in the proband.

**Table 1 genes-14-01587-t001:** Reported cases of LGMD with neurodevelopmental features.

	Bögershausen et al., 2013 [16]		Cabrera-Serrano et al., 2015 [17]		Haberlova et al., 2014 [18]	This Study
Gene	*TRAPPC11*		*GMPPB*		*POMT1*	*SGCA*
Variant	c.2938G>A homozygous	c.1287+5G>A homozygous	c.79G > C/c.1036C>A	c.860G>A /c.458 C>T	c.251G>A homozygous	c.229C>T homozygous
No. of cases	3	5	2	1	2	1
Gender (*n*)	F (3)	M (4), F (1)	M (1), F (1)	M	F (2)	M
Age of onset: muscle weakness (years)	Early school age	Not reported	Early 20s	13	Early 30s	3
Muscular symptoms (*n*)	Proximal muscle weakness, myalgia (3)	Mild muscle weakness (1), hypotonia in early childhood (1)	Proximal limb weakness (2)	Elevated CK	Generalized and limb-girdle muscle weakness (2)	Weakness, elevated CK, positive Gowers’ sign
Developmental delay (*n*)	Motor delay (1)	GDD (5)	-	Speech delay	GDD (1)	Speech delay
Cognitive and Social Features (*n*)	Mild ID (1)	Mild-moderate ID (5)	Autism (1) Learning disability (2)	-	Deteriorating intellectual ability in 30s (2)	Autism
Additional features (*n*)	-	Ataxia, choreiform movements (5)	-	Rhabdomyolysis episodes, epilepsy	Degeneration on brain MRI, psychotic symptoms (2)	-

Abbreviations: LGMD—limb-girdle muscular dystrophies; CK—creatine kinase; GDD—global developmental delay, referring to significant delays in two or more developmental domains. No.— number; M—male; F—female; *n*—number of individuals.

**Table 2 genes-14-01587-t002:** Assessment of the patient’s autistic features at 39 months using the Childhood Autism Rating Scale, Second Edition (CARS-2).

ASD Features	CARS-2 Score
Relating to people	2.5
Imitation	2.5
Emotional response	3.0
Body use	3.0
Object use	3.0
Adaptation to change	3.0
Visual response	2.5
Listening response	3.0
Taste/smell/touch response and use	2.5
Fear or nervousness	2.5
Verbal communication	2.5
Nonverbal communication	2.0
Activity level	2.5
Level and consistency of intellectual response	2.0
General impressions	3.0
Total Score	39.5

The CARS-2 is a behavior rating scale intended to help diagnose autism spectrum disorders. It is composed of 15 categories that are rated as follows: 1 = normal for the child’s age, 2 = mildly abnormal, 3 = moderately abnormal, and 4 = severely abnormal. A total score ≥ 30 is consistent with autism spectrum disorder, as seen in this patient.

## Data Availability

Not applicable.

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
