# Peer review of "Autosomal Recessive Limb-Girdle Muscular Dystrophy-3: A Case Report of a Patient with Autism Spectrum Disorder"

_genes, 2023, doi:10.3390/genes14081587_

Round 1
Reviewer 1 Report
The Authors report a first case of Autosomal Recessive Limb-Girdle Muscular Dystrophy-3 (“LGMDR3”). Interestingly, for the first time a possible linkage between an SGCA gene common missense mutation and severe neurodevelopmental pathologies (autism spectrum disorders) has apparently been observed.
Major points
The genetic data showing the presence of the homozygous c.229C>T mutation in the patient, and ideally also in his parents, should be reported. This should be a condition sine qua non to be considered for publication.
In Table 2 a series of symptoms are reported. Although interesting, this seems a bit sketchy and the tests applied in order to register/record the symptoms are not described at all.
As stated by the Authors, the missense mutation is kind of common, and as such it has been already identified and described by others (Alonso-Perez et al., 2020, quoted in the Ref list). With this missense mutation, Arg77Cys, a significant reduction (<30%) In the levels of alpha-sarcoglycan is commonly observed. Unfortunately, the Authors do not present neither histological nor immunological (Western blot) data, therefore it remains difficult to run a comparison with the degree of severity of symptoms observed in previous cases. Are patient biopsies available? Can some additional experiments be added in order to better assess the aforementioned point?
If possible, it would be also interesting to check the “overall state” of the DGC, at least in the striated muscle. This could be done also checking the other sarcoglycans and ideally the dystroglycan complex too, with a series of commercially available antibodies.
It is unclear whether the variants of STXBP2 and SSR4 genes found through WGS are in heterozygous of homozygous fashion. Probably heterozygous? There are no direct genetic data reported on the two variants found in the STXBP2 gene (one is reported to be pathogenic) and on the one in the SSR4 gene. The variants found should be clearly reported and some references added for the diseases linked to the two genes.
Given the early signs of muscular weakness and neurodevelopmental symptoms recorded, I think that possible contributions to the observed neurodevelopmental pathology of other genetic factors cannot be completely ruled out. I would suggest to add a sort of disclaimer sentence to the final paragraph.
Minor points
_The amino acid “switch” from Arg to Cys should be also reported in the Abstract.
_Please check, Gower’s sign should be changed to Gowers’ sign.
_In Tab.1, “Lewis et al. 2023” should be changed to “this study”.
_All codes for genes should be reported in italics.
Minor editing of English language required.
Author Response
We would like to thank the reviewer for the thoughtful comments and provide responses to the comments.
Major points
- The genetic data showing the presence of the homozygous c.229C>T mutation in the patient, and ideally also in his parents, should be reported. This should be a condition sine qua non to be considered for publication.
Response: We have included images of Sanger sequencing chromatography from the commercial lab.
- In Table 2 a series of symptoms are reported. Although interesting, this seems a bit sketchy and the tests applied in order to register/record the symptoms are not described at all.
Response: We have reformatted the table and added more objective data including the CARS-2 score of this patient. The diagnosis is made by DSM-5 criteria for Autism spectrum disorders which was performed by the developmental pediatrician. The other 2 most sensitive assessments for the diagnosis of autism are ADOS and ADIR, which were not performed for this patient.
- As stated by the Authors, the missense mutation is kind of common, and as such it has been already identified and described by others (Alonso-Perez et al., 2020, quoted in the Ref list). With this missense mutation, Arg77Cys, a significant reduction (<30%) In the levels of alpha-sarcoglycan is commonly observed. Unfortunately, the Authors do not present neither histological nor immunological (Western blot) data, therefore it remains difficult to run a comparison with the degree of severity of symptoms observed in previous cases. Are patient biopsies available? Can some additional experiments be added in order to better assess the aforementioned point? If possible, it would be also interesting to check the “overall state” of the DGC, at least in the striated muscle. This could be done also checking the other sarcoglycans and ideally the dystroglycan complex too, with a series of commercially available antibodies.
Response: Muscle biopsy was not necessary for the diagnosis, given that the mutations found were previously described as pathogenetic. Muscle biopsy is a surgical procedure that requires clinical justification or a research protocol that prevented us to proceed. Although, the reduction of alpha-sarcoglycan can be used as biomarker for clinical severity, it may relate to muscular dysfunction. Muscular alpha-sarcoglycan reduction may not be related to brain expression and function or may not correlate with intellectual developmental disorders. However, a larger study of expression and clinical phenotype will be necessary to make an accurate determination of phenotype and genotype associations. We have added this clarification to the discussion.
- It is unclear whether the variants of STXBP2 and SSR4 genesfound through WGS are in heterozygous of homozygous fashion. Probably heterozygous? There are no direct genetic data reported on the two variants found in the STXBP2 gene (one is reported to be pathogenic) and on the one in the SSR4 The variants found should be clearly reported and some references added for the diseases linked to the two genes.
Response: We have added the following information to the manuscript: The variants STXBP2 were a heterozygous pathogenic variant (c.1247-1 G>C;p.? ) and a heterozygous variant of uncertain significance (c.1286 C>T; p.A429V, respectively). The variants were in trans. Biallelic pathogenic variants in STXBP2 cause autosomal recessive familial hemophagocytic lymphohistiocytosis type 5, which typically manifests with prolonged and high fevers, cytopenias, hepatosplenomegaly and liver dysfunction in early childhood [reference provided, PMID: 20558610]. Additionally, a maternally inherited hemizygous missense variant of uncertain significance was detected in the SSR4 gene (c.7G>T; p.A3S). Loss of function pathogenic variants in SSR4 are associated with X-linked SSR4-Related Congenital Disorders of Glycosylation. All males reported with this disorder had global developmental delays, intellectual disability, microcephaly, and typical facial dysmorphism; seizures and gastrointestinal disorders were also common [reference provided, PMID 33300232]. Our patient did not have the typical facial features or microcephaly seen in these patients. No additional variants associated with developmental delay or autism were identified.
- Given the early signs of muscular weakness and neurodevelopmental symptoms recorded, I think that possible contributions to the observed neurodevelopmental pathology of other genetic factors cannot be completely ruled out. I would suggest to add a sort of disclaimer sentence to the final paragraph.
Response: a disclaimer sentence was added.
Minor points
_The amino acid “switch” from Arg to Cys should be also reported in the Abstract.
Response: the change was made.
_Please check, Gower’s sign should be changed to Gowers’ sign.
Response: the change was made.
_In Tab.1, “Lewis et al. 2023” should be changed to “this study”.
Response: the change was made. Note that the table format was changed according to comments from Reviewer #2.
_All codes for genes should be reported in italics.
Response: the change was made.
Reviewer 2 Report
This paper is needed and actual, not so many cases described, an attempt to systematization increases the value for the researchers and clinicians dealing with neuro-degenerative muscular genetic disorders.
Suggestions:
1. Line 17: add that the mutation c.229C>T is in homozygous state in the abstract body
2. Re-arrange Table 1: make lines columns and columns lines to save space and make it more descriptive.
3. Adding more detailed graph of the pedigree would add value.
4. Is there substitutional therapy available – any options? It should be described in the discussion section. There are small molecules, belonging to the class of cystic fibrosis transmembrane regulator (CFTR) correctors, for recovering mutants of α-SG defective in folding and trafficking (Scano M, Benetollo A, Nogara L, Bondì M, Dalla Barba F, Soardi M, Furlan S, Akyurek EE, Caccin P, Carotti M, Sacchetto R, Blaauw B, Sandonà D. CFTR corrector C17 is effective in muscular dystrophy, in vivo proof of concept in LGMDR3. Hum Mol Genet. 2022 Feb 21;31(4):499-509. doi: 10.1093/hmg/ddab260. PMID: 34505136; PMCID: PMC8863415.)
5. There are some gene therapy and base editing approach for SGCA c.157G>A mutation, but not to c.229C>T mutation. I would suggest to add some discussion on this and available options for the mentioned clinical trials enrollment.
6. Line 84.- typo with comma: "children ,and difficulty"
Author Response
We would like to thank reviewer 2 for the positive comments and revisions made and have made all changes suggested.
- Line 17: add that the mutation c.229C>T is in homozygousstate in the abstract body
Response: the change was made.
2. Re-arrange Table 1: make lines columns and columns lines to save space and make it more descriptive.
Response: We tried to make changes in the format of the table which resulted in a larger table, therefore, we have decided to leave it as it was originally submitted.
- Adding more detailed graph of the pedigree would add value.
Response: pedigree added
- Is there substitutional therapy available – any options? It should be described in the discussion section. There are small molecules, belonging to the class of cystic fibrosis transmembrane regulator (CFTR) correctors, for recovering mutants of α-SG defective in folding and trafficking (Scano M, Benetollo A, Nogara L, Bondì M, Dalla Barba F, Soardi M, Furlan S, Akyurek EE, Caccin P, Carotti M, Sacchetto R, Blaauw B, Sandonà D. CFTR corrector C17 is effective in muscular dystrophy, in vivo proof of concept in LGMDR3. Hum Mol Genet. 2022 Feb 21;31(4):499-509. doi: 10.1093/hmg/ddab260. PMID: 34505136; PMCID: PMC8863415.)
- There are some gene therapy and base editing approach for SGCA c.157G>A mutation, but not to c.229C>T mutation. I would suggest to add some discussion on this and available options for the mentioned clinical trials enrollment.
Response to 5 and 6: To our knowledge there are no therapeutic clinical trials at this time, but the patient was referred to a research center for natural history research participation and potentially further clinical evaluation. We also added a small paragraph on potential therapeutics for this condition at the end of the discussion
- Line 84.- typo with comma: "children ,and difficulty"
Response: the change was made.